# Seaweeds Compounds: An Ecosustainable Source of Cosmetic Ingredients?

**Tiago Morais** [1,2] , **João Cotas** [2] , **Diana Pacheco** [2] and **Leonel Pereira** [2,*]

1 Lusalgae, Lda., Incubadora de Empresas da Figueira da Foz, Rua das Acácias No. 40–A, 3090-380 Figueira da Foz, Portugal; tsmorais@lusalgae.pt

2 MARE-Marine and Environmental Sciences Centre, Department of Life Sciences, University of Coimbra, 3001-456 Coimbra, Portugal; jcotas@uc.pt (J.C.); diana.pacheco@uc.pt (D.P.)

* Correspondence: leonel.pereira@uc.pt; Tel.: +351-239-240-782

**Abstract:** Seaweed-based cosmetics are being gradually used by consumers as a substitute of synthetic equivalent products. These seaweed-based products normally contain purified compounds or extracts with several compounds. Several seaweeds' molecules already demonstrated a high potential as a cosmetic active ingredient (such as, mycosporine-like amino acids, fucoidan, pigments, phenolic compounds) or as a key element for the products consistency (agar, alginate, carrageenan). Moreover, seaweeds' compounds present important qualities for cosmetic application, such as low cytotoxicity and low allergens content. However, seaweeds' biochemical profile can be variable, and the extraction methods can cause the loss of some of the biomolecules. This review gives a general look at the seaweed cosmetics benefits and its current application in the cosmetic industry. Moreover, it focuses on the ecological and sustainable scope of seaweed exploitation to guarantee a safe source of ingredients for the cosmetic industry and consumers.

**Keywords:** seaweeds; cosmeceutical; seaweed ecology; sustainability; seaweed compounds



## 1. Introduction

Seaweeds (macroalgae) are photoautotrophic aquatic organisms (they use photosynthesis to metabolize the necessary energy to carry out their physiological processes), which inhabit almost every corner of the planet earth, especially coastal areas. Due to their trophic level they are considered as primary producers, similar to terrestrial plants. Although seaweeds and terrestrial plants have some common traits, such as the presence of chlorophyll, algae are non-vascular organisms and do not share some physiological characteristics with terrestrial plants, such as a root, stem and leaf division or conductive vessels [1].

From a taxonomic perspective, the easiest way of addressing the different seaweeds is to group them into supergroups defined by the coloration presented by the seaweed thallus (green, red and browns seaweeds) [1,2].

Seaweeds biotechnological potential is diversified, and they can be directly or indirectly employed in different industries, such as pharmaceutical, food and feed, agricultural, bioenergy and cosmetics [1,3–6]. Within the worldwide economy, cosmetics business is among the foremost growing and profitable industries. In agreement with some reports, each woman spends around $15,000 in cosmetic products in her lifetime [7,8]. According to Eurostat, the cosmetics market has foreseen an annual total income of US $170 billion [8,9]. In 2016, the European cosmetics market was esteemed at €77 billion (wholesale rate), followed by the North and South American countries, namely USA and Brazil [7,8]. In several developing countries, the increase of the middle-class population will allow the continuously growing of beauty products market [10]. This econometric analysis suggests that the cosmetic industry is a market in expansion, where the innovation is also prioritized, leading to the research and quest for novel products. Nowadays, most of the cosmetic products available in the market are composed by synthetic chemicals, which can result

in an expensive product and lead to noxious side effects for the consumer [8]. Within the increase of consumer awareness regarding these disadvantages, natural based products are preferred among the clients [1,8].

It was discovered that several primary and secondary seaweeds' metabolites had skin care potential, as one of the true multiple applications. Some of those metabolites have the ability to supply moisture to the skin, increase blood flow and enhance cellular renewal [1,8]. These compounds can also interfere in other metabolic processes of the cell, such as the increase of cellular resistance, sebaceous glands regulation and tissue drainage [1,11].

One of the fastest growing cosmetic industries is the cosmeceutical industry, in which topical cosmetic-pharmaceutical hybrids with bioactive compounds in their constitution presents skin health benefits. In recent years, due to the modern lifestyle and ineffectiveness of synthetic cosmetics, the cosmeceutical industry is shifting to natural bioactive compounds [3]. The referred synthetic cosmetics ineffectiveness includes disadvantages and harmful side effects, such as irritation or allergic reactions, and low absorption rate [3]. Theoretically, the skin is only able to absorb molecules with a certain molecular weight (<500 Dalton). Bos and Marcus [12] used cyclosporin, a topical immunosuppressant, with a molecular weight of 1202 Da to assert this. They applied it on psoriasis cases in two different ways: by injecting it directly into the skin or through topic application. The experiment results showed that only the injection was working, and this was due to the high molecular weight of the compounds which inhibits skin penetration if topically applied [3,12].

Because of findings such as these, there is an urgent need to replace synthetic cosmetic ingredients with natural ones, such as seaweed bioactive compounds, which have pharmaceutical potential and proved low cytotoxicity in human cells [3,13].

For instance, it is not strange that the North American competent regulatory authority, Food and Drug Administration, authorized the application of phycocyanin, usually produced by red seaweeds and cyanobacteria, as a natural color addictive in food and cosmetic industries due to its non-toxic, natural and biodegradable characteristics [3].

Another example includes astaxanthin, a carotenoid, which is a free radical scavenger with a high antioxidant activity. Thus, the application of this pigment in food market, as a dietary supplement has aroused interest among some of the cosmeceutical industry key players, namely the companies Unilever (United Kingdom), L'Oreal (France) and the German enterprises Henkel and Beiersdorf [3,14].

The chemical diversity and unique properties that seaweeds present are the reason why they have been subject of interest for the past few years in cosmetic industries. Seaweeds protein, lipidic, phenolic and carbohydrate profile presents cosmetic and cosmeceutical potential, as well as pigments, vitamins and other macro and microelements [1,3,4].

This review does an overall view of the seaweeds' exploitation for cosmetic purposes in the recent years, in which biotechnological advances allow us to incorporate seaweeds' metabolites into cosmeceutical products. The next steps for sustainable and safe exploitation of seaweed for the cosmetic industry are also analyzed.

## 2. Seaweed Biology

Seaweeds are multicellular, macroscopic, eukaryotic, autotrophic, and benthic (at least some species) organisms ubiquitously distributed along all kinds of coasts from tropical to polar regions. As said before, seaweeds' got a lot in common with land plants, consequently, they are in the basis of marine ecosystem food web [5]. Marine algae are considered the most important primary producers, which are comprised by photosynthetic organisms (macroalgae or benthonic algae) that inhabits the intertidal and sub-tidal regions of coastal areas [5].

Nowadays, it is academically consensual the seaweeds division in two kingdoms and three phyla: green algae (Chlorophyta phylum) and red algae (Rhodophyta phylum) are part of the Plantae kingdom, while brown algae (Ochrophyta phylum, Phaeophyceae class) are a part of the Chromista kingdom. As explained before, this organization is largely

based on the color of the thallus and according to the cellular organization, photosynthetic pigments, reserve substances and cellular wall components as presented in Table 1 [1,5].

**Table 1.** Seaweed classification according to cellular organization, photosynthetic pigments, reserve substances and cell wall components [1,15,16].

| Phylum | Cellular Organization | Photosynthetic Pigments | Reserve SubStances | Cell Wall Components |
|---|---|---|---|---|
| Chlorophyta (Green Seaweed) | Unicellular or Multicellular | Chlorophyll a; Chlorophyll b; β-carotenoids; Xanthophylls. | Starch | Cellulose and Pectin |
| Rhodophyta (Red Seaweed) | Multicellular | R-phycoerythrin; R-phycocyanin; Chlorophyll a; Chlorophyll d; Xanthophylls; β-carotenoids. | Florid Starch | Cellulose, Agar and Carrageenan |
| Ochrophyta (Brown Seaweed) | Multicellular | Fucoxanthin; β-carotenoids; Chlorophyll a; Chlorophyll c. | Laminarin; Starch; Mannitol. | Cellulose and alginic acid |

Proteins, carbohydrates, and lipids are structural biomolecules that seaweeds synthesize, also known as primary metabolites. Concurrently, they also synthesize secondary metabolic products, which possesses a wide range of biotechnological applications for several markets, such as pharmaceutic, alimentary and agriculture [5].

The marine environment is home for many diverse organisms, in which we can find seaweed. Currently, and contrarily to the popular believe that land plants produce the major part of Earth's oxygen, the Ocean is in fact considered the "lungs of the Earth" [5]. Marine algae produce around 80% of the oxygen present in the atmosphere [5].

As an energy source, all of them accumulate starch in their cells, but other polysaccharides of large molecular weight, which differ depending on the division.

The green algae (Chlorophyta) produce the polysaccharide ulvan and contain pigments, such as carotene, xanthophylls and chlorophylls a and b (sustaining the theory that green algae preceded terrestrial plants). Red seaweeds are more present in warmer waters, having chlorophylls a and d and carotenoids. However, their coloration is owed to the high concentration of phycoerythrin in their cells. Brown seaweeds are mostly from colder waters, in which fucoxanthin gives the typical color, they also possess chlorophylls a and c and other carotenoids [17].

Their main ecosystem service is to provide food to the herbivores, being the marine food chain basis, responsible to sustain various benthonic communities [18]. In the wild habitat, they dispute the space for the best spot of nutrient concentration, light incidence, and available area, however, it is indispensable water (within a specific salinity range) and carbon dioxide for seaweed survival and development. They synthetize bioactive molecules to protect their space and from herbivores and contaminations, by fungi's, bacteria's or other algae species [19]. Additionally, to survive they are capable of absorbing pollutants, such as heavy metals and toxins, however from species to species this condition varies. Thus, some seaweeds are good for bioremediation, others as environmental quality bioindicators [20,21].

Seaweeds are sessile organisms living in a multifarious and dynamic ecological niche, which can fluctuate very rapidly and in an extreme way due to the biotic and abiotic changes that also, can be very widely variable, thus the seaweeds survival depends on their resilience. The key factors for seaweeds growth are temperature, salinity, light, pollutants, and nutrients concentration. For their survival, seaweeds metabolize a wide range of compounds (primary and secondary metabolic products), in order to cope with

the stressors, they are exposed to. Thus, some compounds are exclusively synthesized by seaweeds [22–24]. However, this environmental aspect can be a problem to industry due to biochemical changes in compounds concentration and/or quality, as several studies demonstrate the fluctuations [25–29]. Nevertheless, the industry continues to exploit seaweed biomass, focusing more in the compounds extraction [30–33].

### 3. The Seaweeds Compounds: Cosmetic Potential?

Cosmetic products are conceived as cleaning and beautifying agents, with the aim of improving the aesthetics of the user, without harmful side effects. Several products, such as creams, lotions, and ointments are also composed by bioactive molecules (i.e., vitamins, minerals, antioxidants) that can promote skin, nails and hair health [14,34,35]. These products can assume a variety of formats, such as simple cream or lotion, or even edible products, such as pills or functional foods with cosmeceutical activity [34,35]. However, since they cannot claim a real therapeutic function, it is important to distinguish these products from cosmetic preparations, which aims skin diseases prevention, for example, a sunscreen is considered a drug that prevents skin diseases due to the solar exposure [34,35]. In order to accommodate cosmetic products that can claim to have biological action, the term "cosmeceutical" was created, being a nomenclature currently used in countless products [1,36].

As discussed earlier, during many years the cosmetic industry used synthetic ingredients. However, due to multiple factors, such as the modern lifestyle and the ineffectiveness of synthetic cosmetics, a consequence from problems such as low absorption rate or harmful side effects, there is a shift towards the use of natural bioactive compounds [3]. As an example of the possible side effects, let us take a look at the case of the parabens (hydroxybenzoic acid esters), broadly employed in synthetic cosmetic products, mimicking the female hormone, estrogen [3,37]. However, this could increase the possibility of the development of malignant melanoma or breast cancer [3,37]. Another example is phthalate, a compound widely used for plastic production, whereas it could be found on several cosmetic products (i.e., nail polish) and could lead to DNA modification and damage, as was proved in human sperm cells [3,38]. In short, the understanding of skin growth and damage mechanisms, combined with the successful use of natural products as a solution to the problems mentioned before, highlighted the benefits of using cosmetics in a cosmeceutical logic, rather than in a strictly esthetic one [39].

Cosmetics played a central role, since immemorial times, in human society, mainly for religious and ornamental purposes. In ancient times, such products were extracted from natural compounds (i.e., milk, flowers, fruits, seeds, vegetables) and minerals (i.e., clay, ash) [39]. In certain parts of the world, there is a practice of using seaweed as an alternative remedy for skin-related diseases, making it into an incredible natural raw material for cosmetics [1,8]. The bioactive compounds found on seaweed have multiple activities, which allows them to be used as an active ingredient when formulating cosmetics [8,14,40]. Various reviews and research articles demonstrate the wide range bioactivities that seaweeds can offer, such as the prevention of tumors or allergies development, microbial growth inhibition, as well as antioxidant and anti-lipidemic properties [8,39,41–44].

Trending alongside the search for raw materials to cosmetic and cosmeceutical in macroalgae, there is also a recent growing interest in bioactive compounds from microalgae. This subject will not be a part of this review, but it is worth mentioning that the scientific community show us that microalgae have much of the bioactive potential of seaweed and are able to include most of the same applications, including the cosmetic and cosmeceutical products [45].

The application of seaweed to the cosmetic industry is based on their valuable bioactive compounds, such as carbohydrates, phenolic molecules, natural pigments, sterols, proteins, lipids (polyunsaturated fatty acids) peptides, amino acids, vitamins and minerals, and their use as active ingredients due to their potent bioactivity [8,41,46]. We go on to describe the bioactive potential and the biochemical characterization of such compounds.

### 3.1. Phenolic Compounds

Phenols are seaweeds' secondary metabolic products that constitute an interesting group of compounds for cosmetic application, due to their numerous biological activities [8]. It is a diverse group of water-soluble chemical compounds that share a hydroxyl group linked to an aromatic hydrocarbon group. According to the number of substituents, phenols can be categorized into simple phenolic compounds or polyphenols, comprising terpenoids, flavonoids, phlorotannins, bromophenols and several mycosporine-like amino acids [6]. This is one of the major differences between terrestrial origin polyphenols, which are composed of flavonoids and gallic acid [1,8]. Phlorotannins are in higher concentration in brown seaweeds. However, in green and red seaweeds bromophenols, flavonoids, phenolics acids, terpenoids, and mycosporine-like amino acids are usually predominant [6,47,48]. Thus, phenolic compounds can be metabolized with different molecular weights (126 to 65,000 Da) through the metabolic acetate-malonate pathway [1,35].

Phlorotannins acts as an inhibitory agent of hyaluronidase (Haase) activation, conferring to cosmetic products antiallergic, anti-wrinkle, skin antiaging and whitening properties, and since seaweeds are the only producers of these molecules on the whole planet, these turned to be valuable ingredients for the cosmetic industries [1,49].

The bioactivity potential of seaweed phenolics is the consequence of their enzyme inhibitory effect and antimicrobial, antiviral, anticancer, antidiabetic, antioxidant, or anti-inflammatory activities, which can be very attractive to be applied in cosmetic and cosmeceutical products [50].

Although deficiencies in polyphenol intake, this does not result in specific diseases, however the consumption of an adequate level of polyphenols could have health benefits, mainly the prevention of diseases such as obesity, metabolic syndrome, Alzheimer's or cancer [50]. We now discuss the main groups of phenolic compounds.

### 3.1.1. Phenolic Acids

Phenolic acids are bioactive phenols with several functions, such as nutrient absorption, protein synthesis, enzymatic activity, photosynthesis, and allelopathy. Structurally, they are formed by a phenol ring and, at least, a carboxylic acid group, being then classified by the number of carbons present in the chain: hydroxybenzoic acid (if it have only one carbon), acetophenones and phenylacetic acids (if there are two carbons present) and hydroxycinnamic acids (if they have three carbons) [6,48,51–53].

There is some work developed regarding phenolic acids and seaweeds. However, there is still a lack of characterization and bioactivity studies, which turns it impossible to discuss further the theme [6].

### 3.1.2. Phlorotannins

Phlorotannins are oligomeric or polymeric phloroglucinol phenols, found on the algal cell wall, that are involved in the protection of cells from UV radiation [54]. From a biosynthesis perspective, the oligomeric form, phloroglucinol—which is exclusive of brown seaweed—is formed through the acetate-malonate pathway in the Golgi apparatus. Then, they bind through C-C and/or C-O-C forming molecules of polymeric phloroglucinol, ranging from 10 to 100 kDa, which is attributed to the variety of structural bounds within the molecule. Based on that, it is possible to distinguish phlorotannins by six groups: phloretols (aryl–ether bonds), fucols (aryl–aryl bonds), fucophloretols (ether or phenyl linage), eckols (dibenzo-1,4-dioxin linkages), fuhalols (ortho-/para- arranged ether bridges containing an additional hydroxyl group on one unit) and carmalols (dibenzodioxin moiety) [6,55,56]. They can also be divided into linear or branched phlorotannins, since the monomers can bind at any position of the phloroglucinol ring, which leads to the creation of structural isomers, in addition to the conformational isomers explored before [6,55,56].

These compounds attracted a lot of attention due to the multiple activities they present. Between their clinical and cosmeceutical potential antioxidant, anti-inflammatory, antitumor, antiallergic, hyaluronidase inhibitory, and matrix metalloproteinases inhibitory

activities can be counted [54,57]. In addition, as explored before, they show activity not only for UV protection, but also against ionizing radiation. The mechanism responsible for this is the redox activity, which balances the oxidative stress within the cell and reduces the DNA damages [54,58–60]. These characteristics allows the phlorotaninns to modulate the NRF2 pathway, which is linked to both, inflammatory and oxidative stress responses, being an effective natural compound capable of managing inflammatory skin diseases [54,61].

### 3.1.3. Bromophenols

Bromophenols are phenolic compounds, common to all major seaweed group, characterized by the bromination, in various levels, of the phenolic groups. This process happens because the organisms have haloperoxidases capable of halogenating organic substrates in the presence of halide ions and hydrogen peroxide [6,62].

These compounds have, mainly, ecological functions-chemical defense, for example, as an anti-herbivory-but are also responsible for organoleptic seaweed characteristics, such as the flavor [6,62].

Despite the low quantities registered in seaweeds, which presents itself a problem regarding the study of the biosynthesis and other aspects of these compounds [6].

### 3.1.4. Flavonoids

Flavonoids are phenolic compounds characterized by a heterocyclic oxygen linked to two aromatic rings, which vary in the level of hydrogenation. They are widely found in terrestrial plants, but some studies report the detection of catechins and other flavonoids, such as rutin, quercitin and hesperidin, in species from all the three major groups of seaweed [6,63]. However, the studies with these compounds are scarce and contradictory. For instance, Yonekura-Sakakibara et al., 2019, argues that all algae (macro and microalgae) do not contain flavonoids, because they lack two primary enzymes necessary for the flavonoid biosynthesis [64]. Nevertheless, Goiris et al., 2014, had already shown flavones, isoflavones, and flavonols present in various microalgae evolutionary lineages, which can be explained because genes encoding enzymes involved in the shikimate pathway are present in those algae [64–66]. Further studies are needed, not only to clarify this question, but also to explain the biosynthesis of flavonoids [6].

### 3.1.5. Phenolic Terpenoids

Phenolic terpenoids are phenolic compounds who have been detected in brown and red seaweed. Their chemical characterization is diverse. Phenolic terpenoids from brown seaweeds, for instance, have been characterized as meroditerpenoids, which can be divided in plastoquinones, chromanols and chromenes. The chemical structure of these compounds consist of a polyprenyl chain bound to a hydroquinone ring moiety [6,67].

On the other hand, in red seaweed it was recorded the presence of diterpenes and sesquiterpenes [6,68]. Further studies, in order to understand the formation of phenolic terpenoids in seaweeds are needed, since it cannot be certified that these compounds follows the same biosynthesis pathway [6].

### 3.1.6. Mycosporine-Like Aminoacids (MAA)

MAA are UV-absorbing compounds, with low molecular weight (<400 Da) found on the cell cytoplasm. Their chemical structure consists of ciclehexenone or cyclohexenine ring linked to amino acid substituents. The bonds formed between the ring and substituents will then result in a broadband absorption of different wavelengths, depending on the substituents in the structure. Most of these compounds are synthesized by the Shikimate pathway [6,69].

MAA are found not only in seaweeds, but in a wide variety of aquatic organisms. In the case of seaweeds, they are found on red seaweed species and there is some discussion around the detection of MAA in green and brown species [6,69].

### 3.1.7. Non-Typical Phenolic Compound

In some seaweed species, the presence of non-typical phenolic compound was detected. Presenting themselves as Phenylpropanoid derivatives, such as colpols and tichocarpols, they can be found in brown and red seaweed, such as *Colpomenia sinuosa* and *Tichocarpus crinitus* [6,70,71].

It was also detected lignin, a polymerized hydroxicinnamyl alcohol, in the red seaweed *Calliarthron cheilosporioides*, that was thought to be restricted to vascular plants [6,72].

### 3.2. Phycocolloids and Other Polysaccharides

Polysaccharides represent around 60% of all active metabolites occurring in seaweed. They are composed by several building blocks (monosaccharides) linked by glycosidic bonds, forming long chained carbohydrates. These compounds are characterized by being hydrophilic, water-soluble and to possess a regular structure [1].

Polysaccharides have a structural role on seaweeds cell wall and as an energetic reservoir. Thus, seaweeds' contains several polysaccharides with proved biological activities, and can be applied in cosmeceutical products as moisturizers and antioxidant [1,35].

Macroalgal hydrocolloids-known as phycocolloids-are, of all the polysaccharides, the most relevant in terms of their industrial commercialization. Phycocolloids are structural polysaccharides, found in seaweed, that usually form colloidal solutions-an intermediate phase between a solution and a suspension. Hence, polysaccharides can be used in several industries, particularly in cosmetic, as thickeners, gelling and stabilizers agents for suspensions and emulsions [1,35,73]. The main polysaccharides groups are presented below.

### 3.2.1. Agar

Agar, also referred as agar-agar or agarose, is a potent gelatinous hydrocolloid derived from several red seaweed species. This molecule is composed by a heterogeneous mixture of two carbohydrates, agarose and agaropectin, which flows into the cell wall and intercellular spaces as a structural carbohydrate [1,73]. The predominant fraction of agar (50–90%) is agarose, which confers its gelling properties [73,74]. This polysaccharide presents a high molecular weight and it is constituted by several (1–3)-β-D-galactopyranosyl-(1–4)-3,6-anhydro-$\alpha$-L-galactopyranose units, however it has been recorded some modifications on the structure of this molecule, due to several factors, namely the seaweed species, as well as biotic and abiotic parameters variation [1,16]. Agaropectin, a heterogeneous mixture of β-1,3-linked D-galactose containing sulfate and pyruvate moieties, exists in small quantities. The gelling properties of agar depend on the degree of sulphation and the concentration 3,6-anhydrogalactose [73,75].

This polysaccharide is also applied as an emulsifier and stabilizer agent in creams and to regulate the moisture content in cosmetic items, likewise hand lotions and liquid soap, deodorants, foundation, exfoliant, cleanser, shaving cream, facial moisturizer/lotion or for acne and anti-aging treatment [1,76]. Agar can tolerate high temperatures (up to 250 °C), maintain its characteristics even near the boiling-point temperatures, making it suitable for its application in jellied confections, due to the possibility of treating the ingredients at high temperatures and then cool them down [77]. However, there may be other bioactivities in the sulfated-galactan, in which agar is constituted [78,79].

### 3.2.2. Alginic Acid

Alginic Acid is the primary polysaccharide present in many brown seaweed-such as *Ascophyllum*, *Durvillaea*, *Ecklonia*, *Laminaria*, *Lessonia*, *Macrocystis*, *Saccharina*, *Sargassum* and *Turbinaria*—and it is a linear copolymer of β-D-mannuronic acid and $\alpha$-L-guluronic acid units linked by 1→4 glycosidic bonds. The extraction technique is based on the transformation of an insoluble mixture of alginic acid salts into soluble salts, which can be alginate or algin, being appropriated for aqueous extraction [73,80].

In more detail, in the presence of divalent cations such as $Mg^{2+}$, $Ca^{2+}$, $Sr^{2+}$, and $Ba^{2+}$ alginates easily form hydrogels. As a result, the extraction is accomplished through

the conversion of insoluble alginic acid and its salts into water-soluble potassium and sodium alginate. The stiffness or viscosity of these gels will depend on the polymers' structural properties ($\alpha$-L-guluronate residues amount) and it is influenced by the quantity of the external salt [73,79,80]. Furthermore, acidic conditions make it insoluble, producing gel-such as properties with alginic acid's sodium and potassium salts, while retaining water-soluble properties [79].

Alginates are commonly used as gelling agents, as thickeners, protective colloids or emulsion stabilizers, for the formulation of hand jellies and lotions, ointment bases, pomades and other hair preparations, greaseless creams, dentifrices and other cosmetic products because of their chelating properties. This polysaccharide could as well be employed in the formulation of skin protective in prevention of industrial dermatitis. Creams with these ingredients form versatile films of enhanced skin adhesion and are ideal for a range of cosmetics application [73,81].

### 3.2.3. Carrageenan

Some species of red seaweed from several families of the order Gigartinales, known as carragenophytes, produce a pure polysaccharide called "carrageenin", which is an unstable compound and difficult to extract. Therefore, carrageenin binds to one or more cations, producing different carrageenin salts (carrageenan)-which represent something between 30% and 75% of the seaweed dry weight. These linear sulphated polygalactans are composed alternated residues of galactose residues, $\alpha$ (1–3) and $\beta$ (1–4) bonds [1,73,82].

From a commercial point of view, and according to several regulatory authorities (FDA, EFSA), who classified it as safe, since the toxicological evaluation showed minimal or no adverse physiological impact. The most important types of carrageenans' are kappa ($\kappa$), iota ($\iota$) and lambda ($\lambda$). Gelling properties are a characteristic of iota and kappa carrageenan, whereas lambda presents thickening properties [1,82–84]. However, seaweeds, commonly, do not synthesize this idealized carrageenan, while the occurrence of hybrid structures is more usual. For example, theta ($\theta$), xi ($\xi$), beta ($\beta$), mu ($\mu$), and nu ($\nu$) can also occur and when those are exposed to an alkali treatment, they can be modified into kappa and iota, respectively, through the establishment of 3,6-anhydrogalactose bridges [1,73,82–84].

Carrageenan can be extracted from several red seaweeds-such as *Betaphycus gelatinum, Chondrus crispus, Eucheuma denticulatum, Sarcopeltis skottsbergii (formerly Gigartina skottsbergii), Kappaphycus alvarezii, Hypnea musciformis, Mastocarpus stellatus, Mazzaella laminarioides, Sarcothalia crispata*, that are interesting for several industries, being that more than 20% of the carrageenan production is used in pharmacy and cosmetics. Many daily cosmetic items have in its formulation the presence of carrageenan, namely toothpastes, hair wash products, lotions, medicines, sun blockers, shaving creams, deodorant sticks, sprays, and foams [1,79,83,85]. Reports shown the variety of bioactive properties of carrageenan. Their ability to form hydrogels allow them to be used in various areas for their anti-viral, anti-bacterial or even the capacity of managing pathophysiological processes, such as hyperlipidemia. These are strong arguments for the usage of such compounds, considering the reported high safety, effectiveness and biocompatibility, in addition to be biodegradable and non-toxic. Further studies are, however, needed in order to assess the real effectiveness of carrageenan potential [82].

### 3.2.4. Porphyran

A well-studied group of sulfated polysaccharides obtained by aqueous extraction are porphyrans, that could be obtained from *Porphyra/Pyropia* and *Bangia* species (red seaweeds). Porphyran is a linear polysaccharide that presents glycosidic linkages between repeated and alternated units of substituted $\beta$-D-galactopyranose at carbon 3 and $\alpha$-L-galactopyranose units substituted at carbon 4 in a disaccharide arrangement that can be represented as [→3)-$\beta$-D-galactopyranose-(1→4)-$\alpha$-L-galactopyranose-(1→] n [1,86–88].

Studies highlight porphyran applicability for skin whitening, anti-inflammatory, pain reliever and antiulcer bioactivities, thus being interesting for cosmeceutical applications [89].

### 3.2.5. Laminaran

A small glucan that can be present in either soluble or insoluble form is laminaran, also known as laminarin or leucosin. In cold water, the first form is fully soluble, while the second is only soluble in hot water. This polysaccharide consists of β-(1,3)-linked D-glucose with intra-chain branching of β-(1,6) [17,73,90].

This polysaccharide can be found predominantly in seaweeds belonging to the class Phaeophyceae (brown algae), such as *Laminaria, Saccharina*, and, in lower quantities, in *Ascophyllum, Fucus, Sargassum*, and *Undaria*. Laminarans are studied for their anti-tumoral, anti-inflammatory, anti-coagulant, anti-viral, as well as their antioxidant capacities [19]. In cosmetics, laminarin are usually used in anticellulite cosmetics products [73,81].

### 3.2.6. Fucoidan

Fucoidan is a hydrocolloid mainly composed by α-L-fucopyranose residues. However, this sulfated polysaccharide unveils a heterogenous and dynamic structure and composition. Besides α-L-fucose and sulfates, this molecule can also contain other monosaccharides, such as galactose, xylose, mannose, rhamnose, glucose, and/or uronic acids, or even acetylated groups. Moreover, it can also be differences among the molecular weight, branching and substitutions according to the targeted species, as well as the selected extraction and purification method [1,91].

These sulfated polysaccharides are interesting, due to the wide range of bioactivities they present, such as the prevention of obesity, diabetes and tumors development, as well as their anti-coagulant, anti-thrombotic, anti-inflammatory, UV blocker, tyrosinase inhibitor, antibacterial, antioxidative, antihyperlipidemic activities [92–95].

Fucoidans' anticoagulant bioactivity is highlighted by several authors, claiming that this polysaccharide extracted from the seaweeds *Eklonia cava* and *Fucus vesiculosus* has the ability to inhibit thrombin, which is mediated by anti-thrombin III plasma, mimicking the heparin activity [90,96–98].

In general, studies on the anticoagulant/anti-thrombin activity of fucoidans reported that they are directly dependent on the molecular weight, concentration and/or placement of polysaccharide sulfate groups. Furthermore, binding and branching forms and their monomeric composition can also play an essential role in modulating the biological properties of these sulfated polysaccharide [90,94,97].

In fact, Moon et al. [99] found that, when the skin is exposed to UVB (ultraviolet B) radiation, fucoidan can promote the synthesis of procollagen type I and the inhibition of the expression of metalloproteinase matrix. So, it is theorized that this polysaccharide can be employed as a therapeutical agent, in order to prevent skin photoaging. In addition, studies have shown that fucoidans application in human leukocytes can decrease elastase activity, protecting the elastic fibers of the skin. Fucoidans also act as tyrosinase inhibitors and can, as well, minimize skin pigmentation [99–101].

Fucoidan can also protects the hair and skin by eliminating free radicals, reducing inflammation, wrinkles, allergies, and sensitive skin reaction. This polysaccharide can also promote skin elasticity, firmness and brightness, as well as hair safety, growth, rigidity, cleanliness and gloss [102].

The effects of fucoidan rely on a variety of cellular and molecular mechanisms, such as radical scavenging, down-regulation of COX-1/2, MAPK p38, inhibition of hyaluronidase, DPP-IV and extension of APTT and TT. Literature shows that molecular weight, sulfate and fucose content, and polyphenols may have a role to these activities [103–105].

### 3.2.7. Ulvan

The hydrocolloid that can be extracted from green seaweeds is ulvan, which can represent around 8% to 29% of the algae dry weight. The composition of this polysaccharide consists in rhamnose, xylose, glucose mannose, galactose, and uronic acids, grouped into two main repetitive disaccharides, the ulvanobiuronic acids type A [($\rightarrow$4)-β-D-GlcA-(1$\rightarrow$4)-α-L-Rha 3S-(1$\rightarrow$)], and type B [($\rightarrow$4)-α-L-IdoA(1$\rightarrow$4)-α-L-Rha 3S-(1$\rightarrow$)]. This polysaccharide structure, however, may be more variable, due to taxonomic and/or ecophysiological variations [78,106].

In the presence of divalent cations, such as $Ca^{2+}$, $Cu^{2+}$, and $Zn^{2+}$, ulvan has gelling properties in a pH range of 7.5 to 8.0, and tolerates temperatures up to 180 °C. Both rheological and biofunctional properties of ulvans makes them attractive for cosmeceuticals as raw material [1,78,107].

Gel formation by ulvans is a complex mechanism that involves the formation of spherically shaped structures of ulvans in the presence of boric acid and calcium ions. Ulvans have moisturizing, protective, antitumor, and antioxidative properties [78,80,81, 106,108].

Nowadays, due to the described characteristics, ulvan has been applied as a clouding and flavoring agents in beverages and as a stabilizer in cosmetics [109,110]. In addition, the antioxidant activity of ulvan is subject of interest for the cosmetic industries since it was demonstrated in vitro the protective capability against hydrogen peroxide-induced oxidative stress. Furthermore, the presence of glucuronic acid, which have moisturizing properties, and of rhamnosyl residues, studied for their cell proliferation and collagen synthesis capacities, turn ulvan in interesting raw material for the cosmetic industries [110].

### 3.3. Pigments

The coloring of food and cosmetics can be accomplished by the presence of substances derived from natural sources, which also have beneficial effects on human health by serving to protect the body and prevent diseases. Seaweed has several photosynthetic pigments with that capacity divided into three main classes: chlorophylls, carotenoids (carotenes and xanthophylls), and phycobilins [1], which are explained below.

### 3.3.1. Chlorophylls

Chlorophylls are pigments found in every seaweed species, no matter the color they present. This pigment it is found in the interior of the chloroplasts and gives a green color to the plants. Thus, the seaweeds' color is in fact, a result of the mixture of the several pigments that they contain, in which the major pigment defines the phyla that they belong (Chlorophyta, Rhodophyta or Phaeophyceae) [1].

In seaweeds, chlorophylls usually appear in higher concentrations than other pigments. They are used, particularly because of their antioxidant and antimutagenic effects, in the food, pharmaceutical and cosmetic industries [111]. In the red and blue spectrum areas, chlorophyll absorbs light, emitting a green color and thereby having the capacity to be used as natural coloring product [112]. In addition, chlorophylls have deodorizing and antibacterial bioactivities. The antioxidant activity of these compounds and the potential to promote tissue synthesis make chlorophylls an attractive raw material for cosmetic industry application [113,114].

### 3.3.2. Carotenoids

Carotenoids are lipophilic isoprenoid compounds, in which astaxanthin, fucoxanthin, loraxanthin, lutein, violaxanthin, zeaxanthin, are categorized as xanthophylls, whereas α-carotene, β-carotene, lycopene are categorized as carotenes [115]. In general, these compounds present several bioactive properties, such as anti-inflammatory, antioxidant and antitumoral, thus being applied in natural dyes [115–117].

They have pro-vitamin A which can remain active, particularly at low oxygen rates, where normally vitamin C cannot. These pigments can modulate the gene expression

induced by UVA radiation, protecting the skin and eyes from photo-oxidation, and prevent eye disease in humans [118].

Astaxanthin has a wide range of bioactivities, such as antioxidative, anti-inflammatory, antineoplastic, immunomodulatory activity and it is safe to use, turning it in a very interesting ingredient for the cosmetic industry [119,120]. This compound is not only used for its bright red color, which is already applied in the food market, but for their antioxidant bioactivity, that is higher when compared with β-carotene, for example [118,119].

Fucoxanthin is usually found on brown algae and the main responsible for the yellowish-brown or olive-green coloration of these seaweeds. It has high antioxidant activity, and its main bioactivities are antimalarial, antitumor, anti-obesity, anti-inflammatory, antidiabetic, and antiangiogenic activities. It also has protective effects on the brain, skin, liver, blood vessels, bones, and eyes [121,122].

### 3.3.3. Phycobilins

Phycobilins are pigments that are water soluble and present in the cytoplasm or in the cell stroma. When they form covalent bonds with each other, they form complexes known as phycobiliproteins, determining their color. Phycobilins are, from a chemical perspective, an open chain tetrapyrrole chromophores bearing A, B, C and D rings. At preserved positions, these chromophores bind to the polypeptide chain either by one cysteinyl thioester connection through the vinyl substituent on pyrrole ring A or, often, by two cysteinyl thioester connections on both A and D pyrrole rings through the vinyl substituent [73,123]. These pigments are mainly divided into three groups: phycoerythrin, phycocyanin and allophycocyanin [1,124].

R-phycoerythrin is a phycobilin, commonly found algae that belong to the phylum Rodophyta, and can reach a level of circa 0.2% DW in *Polysiphonia stricta* species and *Neopyropia (Porphyra) yezoensis*, 12% in *Palmaria palmata* and *Gracilaria gracilis* and 0.5% in *G. tikvahiae* [73,125,126]. This pigment, together with other phycobiliproteins, has long been applied in food products, as a natural dye [73,127,128]. as well as in cosmetic and pharmaceutical products. These natural colorants are, generally, stable compounds that can tolerate high temperatures, pH variations and light exposure [73,128].

Pigments such as, C-phycocyanin, R- and B-phycoerythrin are, nowadays, employed in several daily cosmetic products, namely lipsticks or eyeliners [73,126].

Antioxidant, anti-inflammatory, neuro-protective, immunomodulatory, antiviral, antitumor, cardiovascular, and liver protection include the biological properties of phycobilins [73,126,128,129].

### *3.4. Lipids*

One of the three main nutrient groups is lipids. They represent the "building blocks" of all living cells alongside proteins and carbohydrates. They are a diversified group of lipophilic organic compounds found in animals, microorganisms, and plants. As a general characteristic, lipids reflect a group of chemical compounds that have a lipophilic character. Seaweeds presents as lipids fatty acids, sterols, glycolipids, phospholipids, and others [117,130,131]. However, algae fatty acids have been studied for its anti-allergic, antioxidant and anti-inflammatory activities. Lipids may also serve as emollient-softening compounds that safeguard the skin from water loss [116].

### 3.4.1. Polyunsaturated Fatty Acids

Seaweeds have a low amount of lipids that turn them into interesting organisms from a nutritional standpoint. The presence of omega-3 and omega-6 polyunsaturated fatty acids (PUFAs) is responsible for this, as marine algae have a higher PUFA content than terrestrial plants. PUFAs are important regulators of several cellular processes as fundamental elements of all cell membranes and eicosanoid precursors [132,133]. They play an important role in health as mediators [132,134]. In many macroalgae species, the

most representative PUFA-reaching, in some cases, 50% of the total fatty acids is frequently the eicosapentaenoic acid (EPA; C20:5n-3) [132,135].

In seaweed, the dominant omega-3 fatty acids are EPA (20:5) and docosahexaenoic acid (DHA; 22:6), which turn algae into a viable option for inclusion in the everyday diet [136–138]. PUFAs are important for human health because they have different bioactivities, such as cardiovascular and coronary defense, anti-inflammatory activity, arteriosclerosis prevention, among others [136,139,140].

PUFAs present interesting features when compared at a seaweed phyla level. In short, *Chlorophyta* species presents higher quantities of C18-PUFAs than C20-PUFAs, while *Rhodophyta* species showed the opposite. On the other hand, *Phaeophyta* species, on the other hand, showed a profile of C18-PUFAs profile next to green algae and a C20-PUFAs profile identical to red algae. Green algae have higher DHA levels, while brown and red algae contain predominantly EPA and arachidonic acid [132,135].

### 3.4.2. Sterols

Because of their lipophilicity, sterols are a category of seaweed compounds included in the lipid class often found on seaweed. Their chemical composition varies from species to species and is synthesized by isoprene unit coupling, the same route to the diterpenes, but is somewhat different from that of the other lipids and is thus known as secondary metabolites [141].

The most reported sterol in brown algae is fucosterol and can be found around 0.7 to 2.3 g/kg (dry weight) in certain *Undaria* and *Laminaria* species, moreover this sterol has several benefits for human health, such as antioxidant, antiphotoaging, anti-atopic, and anti-inflammatory activity. In the seaweed *Sargassum horneri* (Phaeophyceae), fucoesterol can be found up to 16.2% of the total lipids [141–143]. In red algae, cholesterol is the predominant sterol, while green algae primarily have isofucosterol and cholesterol [141,144].

Seaweeds often produce small amounts of a particular form of sterol, sulfated sterols, which have drawn a great deal of attention from pharmacological activities, such as strong anticancer and antibacterial effects, also against resistant strains. For example, capisterones A and B synthesized and isolated from *Penicillus capitatus* (Chlorophyta) increased the activity of fluconazole in resistant strains [132,145,146].

### 3.5. Proteins, Peptides and Amino Acids

Biological macromolecules consisting of one or more amino acid chains are proteins. In all living organisms, they are present and are part of nearly all cellular processes, performing a wide variety of body functions, such as DNA replication, trigger response, and molecular transport. Most proteins are enzymes that catalyze biochemical responses that are important for several metabolic pathways [1].

Proteins are present in a variety of forms in seaweed, such as single or conjugate form, and can contain protein derivatives as well as free amino acids, such as enzymes or peptides [1]. The proteins' chemical structure and where it can be found in the cell will determine the bioactivity. For instance, these biomolecules present several bioactivities with cosmetic application, such as the prevention of tumors development and inflammation, antioxidant, anti-aging and skin proactive. Thus, seaweed proteins can be employed as moisturizers to hair and body and successfully act as a cosmeceutical [1].

In cosmetic products, amino acids usually function as a hydrating agent since many of them are components of the natural moisturizing factor (NMF) in human skin [81]. Algae are a good source of non-essential amino acids, such as alanine, serine, proline, but also of essential amino acids, such as histidine, tyrosine, and tryptophan [116]. *Ulva australis,* for example, presents an antioxidant and antihypertensive bioactivity, due to its constitution in histidine and taurine [147]. On the other hand, researchers pointed that the red seaweed *Palmaria palmata* (Dulse) and the brown seaweed *Himanthalia elongata* (Sea Spaghetti) are rich in serine, alanine, and glutamic acid [148].

Mycosporine-like amino acids (MAAs) are secondary metabolic products that plays an important role in the absorption of sunlight and have protective function of marine organisms from UV radiation [108]. Moreover, seaweed content of MAAs is higher in the summer and they have been detected in diverse organisms, (especially those who inhabit the moderate depth of 0 to 1 m), mainly in red seaweed such as *Chondrus crispus*, *Palmaria palmata*, *Gelidium* spp., *Porphyra/Pyropia/Neopyropia* spp., *Crassiphycus corneus* (formerly *Gracilaria cornea*), *Asparagopsis armata*, *Solieria chordalis*, *Grateloupia lanceola* and *Curdiea racovitzae*. MAAs can actually be used as UV protectors and cell proliferation stimulators in cosmetics and personal care products [149].

*3.6. Other Seaweeds Compounds: Vitamins and Minerals*

Several compounds, namely vitamin A (β-carotene and other carotenoids), a complex of B vitamins (B1, B2, B3, B5, B6, B7 and B12), vitamin C (ascorbic acid), vitamin D and vitamin E (α-tocopherol) can be found in seaweeds [81,150]. Vitamins actuates in vital metabolic paths as an enzyme co-factor precursor [124].

Moreover, due to the permeability of the algal cellular wall regarding low molecular weight constituents, seaweeds nutritional profile can contain trace elements, such as zinc (Zn), aluminum (Al) magnesium (Mg), silica (Si) copper (Cu), iodine (I), selenium (Sn) iron (Fe) and manganese (Mn), and micronutrients such as calcium (Ca), sodium (Na), phosphorus (P), potassium (K) and chlorine (Cl), depending on the environment which they inhabit [81,151]. Minerals have an essential role in vital metabolic pathways as cofactors of different catalytic metalloenzymes [133].

Seaweed are a natural feedstock of bioactive compounds with several benefic effects for the human skin [152]. Iodine, for example, is a key element for hormones related with the thyroid, helping in the metabolism regulation [153–155]. Worldwide, there is a major public health problem derived insufficient iodine intake, leading to iodine deficiency disorders (IDD) [153–155]. To achieve their optimal set, health authorities suggest an intake of 150–299 μg/ day in adults [153]. Typically, iodine reaches the human body by diet, but it can be obtained by other pathways, such as drinking water, breathing of gaseous iodine in the air and through the skin [156].

## 4. Seaweeds in Cosmetics Nowadays

Worldwide, the cosmetic industry growth is rising owed by the new trend and mindset of modern lifestyle, mostly the Westernized world. In recent decades, the cosmetic industry has gradually fluctuated to the commercial incorporation of natural ingredients (bioactive or secondary elements to give consistency to the product) because of the ineffectiveness of synthetic cosmetics. Moreover, the mindset change to privilege natural products over the synthetic cosmetics [151].

Several reports on cosmetic products with synthetic ingredients highlight the lack of safety data and the increase of adverse effects as one of the principal problems. In some cases, these chemicals when tested in vitro and in vivo provoked adverse effects which were not reported before the application in cosmetic products, creating a health risks for the consumers [38]. Thus, modifying the consumers awareness regarding the natural cosmetic products [3], however, there is the case of the emergence of non-tested cosmetic products.

So, the cosmetic industry is continuous researching for innovative and safe ingredients to develop new products with safe sheets protecting the consumers. In this case, the cosmetic industry is developing diverse cosmetic products based on natural ingredients, where seaweed compounds/extracts are used in the cosmetic formulas [157]. Seaweeds, even as a natural source, can always cause adverse effects in the human organism. Thus, none of the cosmetics are perfectly safe for all human consumers [3]. However, natural ingredients are less harmful when compared to synthetic compounds (where the synthesis can result in harmful derivatives, and high chemical reactivity) [3].

Many researchers and the cosmetic industry have become highly involved in the transition from synthetic compounds to natural ingredients, because seaweed compounds

have a broad range of bioactivities with minimal cytotoxicity effects in human cells [13]. Due to seaweeds properties and advantages, the seaweeds are used in the cosmetic industry as biological active ingredients, organic dyes, texturing stabilizers or emulsifiers and as a feedstock to various interesting molecules applied in skincare (Table 2) [132]. Moreover, seaweeds are photosynthetic organisms, thus they produce secondary metabolites which protect the cell mechanisms and organelles against UV rays, such as pigments and phenolic compounds, which can be used as photo-protective formulation ingredient to produce sunscreens [158].

**Table 2.** Cosmetic properties and applications of seaweed compounds.

| Seaweed | Compound | Properties and Technical Function | Ref |
|---|---|---|---|
| **Brown Seaweed** | Alginate | Emulsifier, viscosifying, moisturizing, chelating, colloids, gelling, immunostimulating, protective colloid agent | [132,159–161] |
| | Fucoidan | Antioxidant, anti-cellulite, antiviral, anti-inflammatory, antiaging, anti-photoaging, elastase, tyrosinase inhibitor agent | [101,162] |
| | Laminaran | Antioxidant, anti-cellulite, anti-inflammatory agent | [116] |
| | Pigments | Antioxidant, UV protector, colorant, and dye agent | [148,158,163] |
| | Phlorotannins | Antioxidant, Collagen-producing enhancer, anti-inflammatory, antioxidants, antiaging, anti-photoaging, anti-allergic, chelating, UV protection, anti-histaminic, anti-wrinkling, hair growth promoter agent | [49,100,116,117,164–166] |
| **Red Seaweed** | Carrageenan | Thickening, viscosifying, stabilizer, sensory enhancer, moisturizing, Anticoagulant, antinociceptive and anti-inflammatory agent | [132,159,167–169] |
| | Agar | Emulsion stabilizer, gelling, thickening agent | [78] |
| | Pigments | Antioxidant, colorant, dye, antioxidant, anti-inflammatory, radical scavenging agent | [116,117,148,158,163,170] |
| | Mycosporine-like amino acids (MAAs) | UV protection agent | [132,148,171,172] |
| | Bromophenols | Antioxidant, antimicrobial, antithrombotic agent | [173] |
| | Fatty acids | emollients, antimicrobial, antioxidant, regenerating compounds, anti-allergic, antiaging, anti-inflammatory, antiaging, anti-wrinkle agent | [130,174] |
| | Sterols | Anti-allergic, anti-inflammatory, antioxidant, radical scavenger agent | [116,117,175] |
| **Green seaweed** | Ulvan | Antioxidant, chelating, gelling, moisturizing, protective agent | [106,176] |
| | Chlorophylls | Antibacterial, antioxidant, antibacterial, deodorizing, tissue growth stimulating, colorant agent | [100,112,113,177] |
| | Carotenoids | Anti-inflammatory, antiaging, antioxidant, tyrosinase inhibitor, anti-photoaging agents, radical scavengers, colorant agent | [100,112,115,117] |
| | Fatty acids | Antioxidant, cytoprotective agent | [178] |
| **All seaweeds** | Amino acids | Antioxidant, antiaging, moisturizing, antioxidant, UV protection agent | [117,132,147] |
| | Proteins | Radical scavengers, antioxidant, chelating, moisturizing, antioxidants, UV protection agent | [117,147,179,180] |
| | Vitamins | Protects against DNA damage. Ability to promote cell regeneration | [181] |
| | Minerals | Minerals reposition | [1] |

As demonstrated in the Table 2, the isolated seaweed compounds demonstrate high potential to be incorporated into cosmetic formula, although extracts and specific compound-enriched extract are already used by the cosmetic industry [1]. Seaweeds are considered an

excellent source of novel natural compounds due to the multiple role effect in the skincare, presenting low cytotoxicity in topical application [7,13,100,182]. Moreover, there are new green extraction methods to obtain purified seaweed extracts with lower environmental negative impacts [81].

However, creating a working formula with seaweed extracts is still difficult due to compounds in the extracts, color, scent and incompatibility with other ingredients in the cosmetic formula [183]. Accordingly, it can be hard to achieve the ideal cosmetic formula with seaweed extract [183]. Consequently, academia and the cosmetic industry are drifting toward specific extracts and compounds, easing the work in cosmetic product formula research. The more the cosmetic chemical researchers know about the compounds/extracts effects, the better the reduction of risks of incompatibilities, color and scent [184], improving the cosmetic product and its stability.

### 4.1. Cosmetic Products Development

From the identification of the bioactive compounds up to the commercialization, there is a need to do biochemical and pharmacological assays to guarantee the active compound and the final cosmetic product safety and properties to be sold.

Although, these cosmetic trials are not public, due to be an aggressive market, and do not have the restriction of the pharmacological industry, which the drugs need to have phase trials to demonstrated efficiency and safety to the living organisms. However, the cosmetic products (even the cosmetic products considered cosmeceuticals) are not considered drugs, as they are not considered as therapeutic, e.g., for treating or preventing diseases or affecting the organism function. Even though the cosmetics affect appearance, the cosmetic is only permitted to be used to clean, beautify, enhance attractiveness or alter the appearance. If a cosmetic products does more than this, it is considered a drug and pharmaceutical product, which has more restrictions [185].

However, the cosmetic products need to have experimental data to certify the performance of the cosmetic product. At this moment, the animal's trials are being prohibited and there are new methods to do cosmetic assays, at this moment with in vitro studies with human cells to understand the cosmetic safety, dosage, and benefits effects. In the case where the cosmetic claim-proving data needs to be inserted into the cosmetic product design, it should be accurately done, showing the specific benefit to the skin. For example, in the European Union and in the United States, the cosmetic products need to have assays providing the benefits before the products commercialization and communication (nowadays, the product marketing is restricted to scientific data, to secure the consumers safety and confidence) [186,187].

As a natural resource, seaweeds can be inserted into cosmetics as plant extract or in an isolated compound, where the compound isolated needs to be evaluated and characterized from pharmacological point of view, to develop the safety requirements.

Seaweed Cosmetic Pharmacokinetics

The pharmacokinetics is an especially important pharmacological branch which is dedicated to analyzing the fate of the substances administered to the living organism. In the cosmetic, these assays occur from the cosmetic/compound application in the specific location of the body until it is eliminated from the body. In this case, the pharmacokinetics will intervene mainly in the dosage of the compound/product to obtain the best results, and to observe when the dosage can have harmful effects. In the cosmetic safety, the pharmacokinetics is very important key to the cosmetic safety sheet [187].

The cosmetic topical application is considered a secure way to apply the cosmetic products, due to the skin barrier which prevent permeation of the most compounds to the human organisms, only compounds with low molecular weight (<350 Da) can be pass the diffusional barrier and enter in the human organisms [188–191].

The cosmetic pharmacokinetics, need to evaluated if the cosmetic compound passes to the human organism and how it can effects and compound half-life in the human organism

until the compounds disappear [192]. In the cosmetics products, the pharmacokinetics is mainly skin-based, such as skin absorption, distribution in the skin, effects in the skin cells metabolism and elimination of the cosmetic by the skin [192]. Nowadays, there are two models to evaluate a cosmetic products from kinetics models, physiologically based pharmacokinetic (use physiological and mechanistic approaches) or in vitro kinetic models (uses mainly cells to assays the substances) [187].

As demonstrated in the Sections 3 and 4 of this review, the seaweeds compounds are all natural and can be absorbed by skin cells and used in the cellular metabolism, where the elimination is efficient and natural. The seaweed and its compounds are considered safe for human nutrition and topical application when within the parameters of the safety analysis [32]. On the contrary, the synthetic compounds that can have adverse effects over long time application. Accordingly, seaweed's potential is under investigation in the pharmaceutical and biomedical industries, e.g., alginate for tissue transplants and phlorotannins as a cardiovascular drug [6,193].

However, in the seaweed area this field is starting to grow now in terms of research, due the cosmetic industry secrets. The results are the main core of the cosmetic product success. However, this area is especially important for cosmetic approval and commercially production.

### 4.2. Commercial Exploitation

Nowadays, there are multiple cosmetics companies using seaweed extracts and compounds in their formulations, as an active agent, or a moisturizer, excipient, gelling, thickening, dyes, pigments, preservatives, additives, aroma or fragrance agents [194].

For example, *Gracilaria* species extracts are integrated into various commercial cosmetics, such as hydrogel soap from Sealaria® (Kfar Hess, Israel), facial mask by Balinique® (Miami, FL, USA), and hydrating cream by Thalasso® (Rosa Graf, Stamford, CT, USA) [195].

On the other hand, it can be also employed by a mixture of diverse seaweeds extracts to produce cosmetic products, for example the Sealgae® (Lusalgae, Figueira da Foz, Portugal) and the Aroma Shoppe Seaweed Gel (Jan Benham Cosmetics, Winschoten, The Netherlands) [196,197].

In the extracts, can be a compound class enriched-extraction, for example, ECKLEXT® BG a phlorotannins enriched-extraction of *Ecklonia cava* subsp. *kurome* (formerly *Ecklonia kurome*) (NOF group, Tokyo, Japan) [198]. The green algae *Ulva lactuca* hydroglycolic extract is the main ingredient of Chlorofiltrat® Ulva HG (CODIF, Saint-Malo, France) [199].

Seaweeds compounds, such as the mycosporine-like amino acids from *Porphyra umbilicalis* (red seaweed) are used in sunscreen products, for example, Helionori® (Gelyma, Marseille, France), Helioguard® 365 (Mibelle Biochemistry, Buchs, Switzerland) and Aethic Sôvée® (produced with Photamin, an extract rich in phenolic compounds, commercialized by AETHIC® in London, UK) [200–203].

Another cosmetic industry segment is the production of seaweeds extracts or purified compounds to be incorporated by the cosmetic producers. In this case, these seaweed-based industries are very specialized to work with seaweed. For example, the Natural Solution (Flemington, NJ, USA) produced two extracts (*Ulva compressa* and *Fucus vesiculosus*) to the cosmetic producers which incorporates in the final product [6]. The *Chondrus crispus* extract enriched in sulphated polysaccharides, Gelcarin® (Dupont Nutrition and Biosciences, Wilmington, DEL, USA), is produced by a specialized company to be used in various cosmetic products as gelling, thickener and stabilizer agent [204].

Seaweeds are already exploited by the cosmetic industry, however, as a natural source (finite resource in the wild), it urges the need to evaluate the ecological impact and sustainability of the business, due to the maintenance of the ecological status quo and cost efficiency at an industrial level.

## 5. Seaweed Recourses Exploitation: Benefits and Problems for the Cosmetic Exploitation

Seaweeds plays a vital role in the growth of the natural cosmetic production, however, there is one problematic to be discussed, "How can we explore a natural source without putting more stress to the environment and natural ecosystems?". Thus, seaweed exploitation, requires a blue economy approach and applied ecology and biotechnology to surpass the problem of seaweeds exploitation. The steps to use seaweed in the cosmetic industry in a sustainable way include: sustainable biomass harvesting; biomass treatment and green extraction methods; quality regulations to certify the extract quality and properties [3].

Thus, it is not a coincidence that the seaweeds which are already incorporated in cosmetic products are also the most produced or harvested globally [5]. Although, the scientific research describes interesting compounds from other seaweeds, there is a need to do a multidisciplinary analysis of the costs and profit of the harvesting/cultivation of new species until the final product, to make decisions based on cost analyses of the development of the value chain [5].

### 5.1. Seaweed Harvesting

The wild seaweed harvesting can be a dangerous activity and can also affect the ecosystem, provoking a negative impact in the environment. Moreover, seaweed compounds can vary greatly in the natural ecosystem due to biotic and abiotic factors, presenting a problem for the seaweed exploitation, and also, there is the issue of the amount of seaweed biomass necessary for the various industries and its growing demand [205]. Thus, the seaweed cultivation can be a solution to control the seaweed quality and reduce the impact in the marine ecosystems [206,207]. The only seaweeds that should be exploited from the wild are the ones who have a fast growth rate on their natural habitat or the ones that present a steady biomass in the marine environment [124].

Seaweed cultivation for cosmetic purposes needs optimal conditions for the seaweed to grow and for the syntheses of the target compounds. Seaweed cultivation is already exploited to produce feedstock for human food, animal feed and compounds extraction (mainly, polysaccharides) [5]. Light intensity, salinity, pH regulation, pollution control, availability of nutrients, presence of carbon dioxide, inorganic carbon, temperature and nutrients are key factors in the cultivation of seaweed for the production of well-quality managed seaweed biomass [157,208]. Furthermore, the type of seaweed cultivation is based on the target species and the ultimate destiny of cultivation. However, at this moment, the seaweed cultivation methodologies are still evolving to meet the best sustainable and stable cultivation systems with more control the seaweed compounds, cost efficient and seaweed production safety [25,30,209,210].

The seaweed quality and productivity can be exploited into an intelligent system (aquaculture 4.0) that can be an advantage, due to the possibility to modification and management of abiotic factors to enhance or reduce seaweed production of the targeted compounds [5]. Which does not happen in wild seaweeds. There are studies supporting these seaweed cultivation advantages, mainly correlating abiotic factors (light quality and intensity) with photoprotector compounds (phenolic compounds and pigments) [211,212]. However, the growth rate is also affected with abiotic variations [212].

### 5.2. Seaweed Biomass Treatment, Extraction Methods

After harvesting the seaweeds, the biomass is directed to biorefinery in order to extract the target compounds and channeled to several cosmetic products [35]. Nevertheless, the manufacture pathway that the seaweed biomass will take will differ, according to the targeted bioactive molecule [81].

Usually, the biorefinery process starts with the algal biomass pre-treatment [1], where the biomass is dried under natural conditions, or in industrial ovens, whereas the temperature does not exceed the 40 °C, to prevent bioactive compounds denaturation, such as proteins, vitamins or enzymes [16].

Alternatively, the biomass can be lyophilized, which is a treatment that is carried at low temperatures and under vacuum. Afterwards, the dried or lyophilized seaweed is grinded until a certain granulometry and it could be directly applied on cosmetic products [8].

However, cosmetic products can be enriched in specific molecules through different extraction methods. For instance, classic methods, such as saponification, maceration, Soxhlet extraction, solid–liquid, and liquid–liquid extraction [3,81]. Moreover, many extraction methods have evolved to obtain a stable, pure and sustainable compound. Thus, extraction techniques assisted by microwave, ultrasound, pulsed electric field or enzymes have been developed, as well as, novel techniques such as, supercritical fluid or water extraction and pressurized liquid extraction [31,213–217]. Other extractions methods which are becoming more researched are deep eutectic solvents that can have less negative impact to the environment and use green solvents, for example, these solvents are equivalent to ionic liquids (molten salts) [218,219]. These green solvents have advantages to be non-inflammable, thermal stability and low volatility [220]. Deep eutectic solvents has been recognized as biodegradable, with lower toxicity and reduced costs than traditional and ionic liquids [221–223]. In the recent research, there was various deep eutectic solvents developed based in the combination of primary metabolites and bio-renewable starting compounds, e.g., sugar alcohols, sugars, amino acids, and organic acids [224,225].

However, it is necessary to consider that the efficiency and quality of these methods can be variable according to solvents, pH and temperature employed, as well as the time range of the extraction [3].

Despite that, the development and selection of low-cost and environmentally friendly bioactive compounds extraction techniques is privileged among the industrial sectors. In this way, is possible to promote eco-sustainable industrial processes, based on a circular and blue economy [31,194,226].

Despite the novel extraction methods, there is a need to characterize the seaweed extract/compound purity and quality.

### 5.3. Quality Check

To guarantee the safety and quality of seaweed-based cosmetics it is necessary to monitor the quality of the algal biomass employed. It urges this need, due to their ability to bioaccumulate organic compounds, such as metals, pesticides or toxins [141,227–229]. For this reason, it is recommended by the World Health Organization that the algal biomass used on cosmetics must be monitored and fulfill a certain threshold regarding the metal concentrations, such as lead, arsenic, mercury and cadmium [81]. Furthermore, it is recommended to do a biochemical profile of seaweed biomass to guarantee biomass quality within the biomass concentration used in the cosmetic formula. The biochemical profile mainly consists in the use of spectroscopic, spectrophotometric and chromatographic methods to characterize the biomass quality and compound concentration (purity) [230,231].

Although there is a uniform will to standardize seaweed-based cosmetic products, there is no legislation regarding seaweed direct or extracts application on cosmetics in Europe [232]. At this point, seaweed extract is considered a vegetal extract, consequently, do not have restrictions in their cosmetic usage [194]. Still, the European Union and the European Free Trade Association holds an Agreement Establishing the European Economic Area (EC-EFTA), which aims to standardize the sampling and monitoring methods of wet or dry algal feedstock, in order to assess a reference point of the algal application on cosmetic products [233]. However, the final product must comply the EU regulation No. 1223/2009, concerning the use of chemical compounds [148]. Where the most important is the chemical safety sheet (with the CAS reference—specific number record in the Chemical Abstracts Service database of a compound) and dosage of the ingredient in the insertion in the EU cosmetic product notification portal [234,235].

Conversely, in the United States of America, there is an entity that regulates cosmetic ingredient safety (Cosmetic Ingredient Review), whereas the compounds are submitted to in vitro and in vivo genotoxicity, carcinogenicity and phototoxicity tests, in order to

evaluate endocrine effects, irritation or sensitization [236,237]. These studies are conducted in order to assess the safety of these compounds to human cosmetic application [238]. Furthermore, there are already approved 82 ingredients of brown seaweeds to cosmetic application in the USA market [236].

Moreover, the Inventory of Existing Cosmetic Ingredients in China (IECIC) reveals that there are 80 algal products allowed in cosmetic products, which are regulated by the National Medical Products Administration (NMPA) [232,239,240].

## 6. Conclusions and Future Insights

Seaweed extracts and purified compounds demonstrate high potential to be incorporated in cosmetic formula, as multiple functions, where can be natural substitutes to the synthetic ingredients.

Moreover, there are diverse companies that already use seaweed extracts and compounds in their formulas. However, the seaweed biochemical profile monitoring presents a problem that seaweed-based cosmetics industries needs to overcome. The development of seaweed cultivation and green extraction methods presents the major keys for this thematic, which is being researched with promising results.

In addition, the creation of new regulations and joint of a multinational cosmetic agency can improve the analytical methods and safe sheets of the seaweed extracts/compounds into the cosmetics products. Enhancing the consumer's safety towards the seaweed-based cosmetics, which is one of the markets that seaweed products are increasing.

**Author Contributions:** Conception and design the idea, T.M., J.C. and L.P.; organization of the team, J.C.; writing and bibliographic research, T.M., J.C. and D.P.; supervision and manuscript revision, L.P. All authors have read and agreed to the published version of the manuscript.

**Funding:** This research received no external funding.

**Institutional Review Board Statement:** Not applicable.

**Informed Consent Statement:** Not applicable.

**Data Availability Statement:** Data sharing not applicable.

**Acknowledgments:** This work was financed by national funds through FCT (Foundation for Science and Technology), I.P., within the scope of the projects UIDB/04292/2020 (MARE, Marine and Environmental Sciences Centre). João Cotas thanks to the European Regional Development Fund through the Interreg Atlantic Area Program, under the project NASPA (EAPA_451/2016). Diana Pacheco thanks to PTDC/BIA-CBI/31144/2017-POCI-01 project -0145-FEDER-031144-MARINE INVADERS, co-financed by the ERDF through POCI (Operational Program Competitiveness and Internationalization) and by the Foundation for Science and Technology (FCT, IP).

**Conflicts of Interest:** The authors declare no conflict of interest.

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
