# Peer review of "Seaweeds Compounds: An Ecosustainable Source of Cosmetic Ingredients?"

_cosmetics, doi:10.3390/cosmetics8010008_

Round 1
Reviewer 1 Report
The work by Morais et al. contains a very comprehensive analysis of the use of seaweed as source of cosmetic ingredients. The topic is interesting and the discussion is sound. I have two very minor commnets before publication
-Grammar and spelling should be revised within the entire manuscript in the current form is difficult to follow the article.
-Authors do not mention chitosans, this is one of the most extended biopolymer categories and there are possibilities for obtaining from different seaweed.
Author Response
Reviewer 1: The work by Morais et al. contains a very comprehensive analysis of the use of seaweed as source of cosmetic ingredients. The topic is interesting, and the discussion is sound. I have two very minor comments before publication.
Answer: The authors acknowledge the kind words from the reviewer, as well as the time spent in order to improve this manuscript.
Comment 1: Grammar and spelling should be revised within the entire manuscript in the current form is difficult to follow the article.
Answer 1: We fully revised the manuscript.
Comment 1: Authors do not mention chitosans, this is one of the most extended biopolymer categories and there are possibilities for obtaining from different seaweed.
https://www.ncbi.nlm.nih.gov/pmc/articles/PMC4557018/
Reviewer 2 Report
L51: and enhance the cell
Table 1 R-phycoerythrin R-phycocyanin
L142: several studies demonstrate
L162: that have been
L412 Omega 3
L431: a group... in seaweeds
L432: varies... and they are
L464: have shown
Table 2: clearly distinguish brown, red and green algae
L533: compounds demonstrate
L544-545: not clear
L604: exploited
L624: extraction methods
L651: methods
L654: it is necessary
L687: future prospects
L688: demonstrate
L729-730: current applications
Author Response
Reviewer 2:
Comment 1:
L51: and enhance the cell
Table 1 R-phycoerythrin R-phycocyanin
L142: several studies demonstrate
L162: that have been
L412 Omega 3
L431: a group... in seaweeds
L432: varies... and they are
L464: have shown
Table 2: clearly distinguish brown, red and green algae
L533: compounds demonstrate
L544-545: not clear
L604: exploited
L624: extraction methods
L651: methods
L654: it is necessary
L687: future prospects
L688: demonstrate
L729-730: current applications
Answer 2: Thank you for pointing these issues. The authors corrected the mentioned typo-errors and reviewed all the text to the text clear.
Reviewer 3 Report
Tiago Morais et al. provide review of Seaweed Compounds as a Source of Cosmetic Ingredients. After close evaluation of the manuscript I recommend revision according to the next points:
- In Introduction: the phrase "...Marine algae are used for different purposes in food, pharmaceutical, energy, agriculture, and cosmetic industries." require support with references.
- The section 2 is not directly suitable for Cosmetic journal. Please consider to remove.
- In Section 3: the phrase "In some areas around the world, there is a tradition in using seaweeds as an alternative medicine for skin-related diseases, turning it in an astonishing natural raw material for cosmetics" require support with corresponding references.
- In Section 3: after the phrase "Many studies revealed the potentiality of seaweeds and associated to their 169 antioxidant, antitumor, anti-inflammatory, anti-lipidemic, anti-microbial, and also their anti-allergic properties" I have expected to see references from "many studies".
- In Scetion 3:I would suggest to discuss next references related to marine phenolics: https://doi.org/10.3390/md18080384; https://doi.org/10.3390/md18100501; https://doi.org/10.3390/md18080377; related to carrageenan: https://doi.org/10.3390/md18110583; related to astaxanthin: https://doi.org/10.3390/md18090459; related to fucoidan: https://doi.org/10.1007/s11094-015-1250-8; https://doi.org/10.3390/md18050275; related to ulvan: https://doi.org/10.1002/ffj.3519; https://doi.org/10.3390/biom10070991; and other reviews and articles: https://doi.org/10.3390/md18060322; https://doi.org/10.3390/md18120644; https://doi.org/10.3390/cosmetics7040087; https://doi.org/10.3390/molecules23102451.
- In Section 5: new extraction methods/solventsincluding natural deep eutectic solvents should be discussed.
- The pharmacokinetics data are essential for understanding of usefulness of seaweed compounds for cosmetic. Please address this issue in the review.
- I would suggest to include special section in which clinically approved cosmetics will be discussed.
Author Response
Reviewer 3:
Tiago Morais et al. provide review of Seaweed Compounds as a Source of Cosmetic Ingredients. After close evaluation of the manuscript I recommend revision according to the next points:
Answer: The authors acknowledge the kind words from the reviewer, as well as the time spent in order to provide a valuable feedback on this manuscript.
Comment 1. In Introduction: the phrase "...Marine algae are used for different purposes in food, pharmaceutical, energy, agriculture, and cosmetic industries." require support with references.
Answer 1: We added more references.
Comment 2. The section 2 is not directly suitable for Cosmetic journal. Please consider removing.
Answer 2: We consider important due the importance and production of metabolites is very important to fully understand the manuscript not only in cosmetic research but from life assessment of the seaweed potential and how to exploit that potential into cosmetic research and development.
Comment 3. In Section 3: the phrase "In some areas around the world, there is a tradition in using seaweeds as an alternative medicine for skin-related diseases, turning it in an astonishing natural raw material for cosmetics" require support with corresponding references.
Answer 3: Thank you, we added the references supporting the sentence.
Comment 4. In Section 3: after the phrase "Many studies revealed the potentiality of seaweeds and associated to their 169 antioxidant, antitumor, anti-inflammatory, anti-lipidemic, anti-microbial, and also their anti-allergic properties" I have expected to see references from "many studies".
Answer 4: We added more supporting references.
Comment 5. In Scetion 3:I would suggest to discuss next references related to marine phenolics: https://doi.org/10.3390/md18080384; https://doi.org/10.3390/md18100501; https://doi.org/10.3390/md18080377; related to carrageenan: https://doi.org/10.3390/md18110583; related to astaxanthin: https://doi.org/10.3390/md18090459; related to fucoidan: https://doi.org/10.1007/s11094-015-1250-8; https://doi.org/10.3390/md18050275; related to ulvan: https://doi.org/10.1002/ffj.3519; https://doi.org/10.3390/biom10070991; and other reviews and articles: https://doi.org/10.3390/md18060322; https://doi.org/10.3390/md18120644; https://doi.org/10.3390/cosmetics7040087; https://doi.org/10.3390/molecules23102451.
Answer 5: We added more information and revised the section 3.
Comment 6. In Section 5: new extraction methods/solvents including natural deep eutectic solvents should be discussed.
Answer 6: We discussed the new extraction methods, thank you for the reviewer advise.
Comment 7. The pharmacokinetics data are essential for understanding of usefulness of seaweed compounds for cosmetic. Please address this issue in the review.
Answer 7: The authors followed the reviewer suggestion and added information about pharmacokinetics in the seaweed cosmteics in the section 4.1.1
Comment 8. I would suggest including special section in which clinically approved cosmetics will be discussed.
Answer 8: Thank you for your suggestion, we did a new section which we discuss this topic (section 4.. At the clinical cosmetics there is a lack of information.
Round 2
Reviewer 3 Report
Authors have implemented my recommendations. The paper was significantly imporvew. It could be accepted for publication.